# Selecting and Interpreting Multiclass Loss and Accuracy Assessment Metrics for Classifications with Class Imbalance: Guidance and Best Practices

Sarah Farhadpour [ID], Timothy A. Warner [ID] and Aaron E. Maxwell *[ID]

Department of Geology and Geography, West Virginia University, Morgantown, WV 26505, USA; sf00039@mix.wvu.edu (S.F.); tim.warner@mail.wvu.edu (T.A.W.)
* Correspondence: aaron.maxwell@mail.wvu.edu

**Abstract:** Evaluating classification accuracy is a key component of the training and validation stages of thematic map production, and the choice of metric has profound implications for both the success of the training process and the reliability of the final accuracy assessment. We explore key considerations in selecting and interpreting loss and assessment metrics in the context of data imbalance, which arises when the classes have unequal proportions within the dataset or landscape being mapped. The challenges involved in calculating single, integrated measures that summarize classification success, especially for datasets with considerable data imbalance, have led to much confusion in the literature. This confusion arises from a range of issues, including a lack of clarity over the redundancy of some accuracy measures, the importance of calculating final accuracy from population-based statistics, the effects of class imbalance on accuracy statistics, and the differing roles of accuracy measures when used for training and final evaluation. In order to characterize classification success at the class level, users typically generate averages from the class-based measures. These averages are sometimes generated at the macro-level, by taking averages of the individual-class statistics, or at the micro-level, by aggregating values within a confusion matrix, and then, calculating the statistic. We show that the micro-averaged producer's accuracy (recall), user's accuracy (precision), and F1-score, as well as weighted macro-averaged statistics where the class prevalences are used as weights, are all equivalent to each other and to the overall accuracy, and thus, are redundant and should be avoided. Our experiment, using a variety of loss metrics for training, suggests that the choice of loss metric is not as complex as it might appear to be, despite the range of choices available, which include cross-entropy (CE), weighted CE, and micro- and macro-Dice. The highest, or close to highest, accuracies in our experiments were obtained by using CE loss for models trained with balanced data, and for models trained with imbalanced data, the highest accuracies were obtained by using weighted CE loss. We recommend that, since weighted CE loss used with balanced training is equivalent to CE, weighted CE loss is a good all-round choice. Although Dice loss is commonly suggested as an alternative to CE loss when classes are imbalanced, micro-averaged Dice is similar to overall accuracy, and thus, is particularly poor for training with imbalanced data. Furthermore, although macro-Dice resulted in models with high accuracy when the training used balanced data, when the training used imbalanced data, the accuracies were lower than for weighted CE. In summary, the significance of this paper lies in its provision of readers with an overview of accuracy and loss metric terminology, insight regarding the redundancy of some measures, and guidance regarding best practices.

**Keywords:** deep learning; accuracy assessment; loss metrics; data imbalance; class-based statistics

## 1. Introduction

The methods used in the assessment of class labeling success are of profound importance in designing the procedures for classification or thematic mapping projects [1,2], though the topic is often given limited attention in the literature. The evaluation of class

labeling success is carried out at two key points in a classification. For methods that rely on the iterative backpropagation of errors and optimization algorithms, such as deep learning-based methods, the choice of loss function, used to quantify whether or not each iteration has reduced classification error, directly affects the resulting classification, since the loss metric acts as the sole indicator for model performance during training and guides parameter updates [3,4]. Second, the final evaluation of the accuracy of a classification is an important attribute in defining the usefulness of the classification model, but if the accuracy estimates themselves are biased, then the value of the model and its products is unclear. In addition, if we wish to compare methods or datasets, biased accuracy measures will likely result in incorrect evaluations [1].

The challenge with choosing accuracy assessment metrics is that we typically want a single metric to facilitate simple yes–no decisions, as in evaluating successive classification iterations with loss metrics, or to facilitate the ranking of classifications in experiments comparing different methods or datasets. Overall accuracy (OA), defined as the probability that a randomly selected sample is correctly classified, superficially seems to fit this need, since it provides a single, integrated metric. However, it is well known that a classification can entirely miss an extremely rare class, and yet have an OA close to 100%. This has led many analysts to claim that OA is misleading in the case of imbalanced (i.e., non-equal) class prevalence. However, this claim is not correct; as pointed out by Stehman and Foody [5], OA is not wrong or misleading, and does not underweight rare classes. The problem is instead that OA is the wrong choice for evaluating the success of discriminating individual classes, as OA merely evaluates whether the label (irrespective of the class) is correct. If the aim is to evaluate classification success on an individual-class basis, a class-based metric is needed. However, using class-based metrics causes its own problems, since defining the fundamental success of labeling at a class level requires, at a minimum for each class, two non-redundant class-based metrics, namely the user's accuracy (UA, also known as precision) and producer's accuracy (PA, also known as recall) [5,6]. Thus, characterizing the accuracy of the individual classes for a three-class classification requires a total of six separate accuracy measures (three UAs and three PAs).

Analysts have attempted to overcome the problem of multiple-class-based metrics by averaging them in various ways. One approach is a simple arithmetic average of the class statistics, known as macro-averaging. An alternative approach is to use micro-averaging, in which the values for the individual classes are first aggregated within a confusion matrix, and then, a single combined statistic is calculated [3,7,8] (the confusion matrix, micro- and macro-averaging, as well as the various accuracy measures, such as UA and PA, are defined and explained in more detail, in Section 2). In choosing a method for combining the class statistics to produce an integrated measure, it is important to consider the implications of these methods, as well as the anticipated use of the derived statistic. For example, are the relative proportions of each class in the reference data meaningful (i.e., do they match the class abundance in the landscape being classified), and is the aim of the classification to produce high OA or to maximize the accuracy of each class or a particular subset of classes, even at the potential expense of OA? These issues are rarely discussed in the literature, adding to the difficulty of designing appropriate experimental methods.

In this paper, we provide a conceptual summary of the major loss metrics used in training and the accuracy assessment metrics used in evaluating classification success, with an emphasis on integrated summary metrics. The scope of this paper includes the exploration of the concept of imbalanced classes, and the implications for both training and classification evaluation. Through a series of experiments, we illustrate the different choices of accuracy measurements and methods for integrating them. The significance of this paper is that it provides guidance and outlines best practices, particularly for implementing macro- and micro-averaging for calculating loss metrics and multiclass accuracy in the context of imbalanced and balanced data.

The rest of this paper is organized into four parts. Following this introduction, Section 2 provides background material regarding accuracy assessment metrics, the mul-

ticlass averaging of class statistics, and loss metrics. Section 3 summarizes the data and methods for the experiments. Section 4 uses the results of the experiments to explore issues related to micro- and macro-averaging for accuracy assessment and loss metrics in the context of class imbalance. The conclusions are presented in Section 5.

## 2. Background

### 2.1. Accuracy Assessment Metrics

More than three decades of research into the assessment of the accuracy of remote sensing products has resulted in an extensive body of literature. Stehman and Czaplewski [9] provide a comprehensive discussion of the basic components of a statistically rigorous accuracy assessment of thematic maps, including sampling design, response design, and estimation and analysis protocols. They stress the importance of using probabilistic sampling designs to ensure a rigorous statistical foundation for inference. Stehman [10] explains that probability-based sampling methods ensure that each element in the map population (e.g., pixel or sampling unit) has a known and non-zero chance of being selected in the sample. He also emphasizes the importance of consistent estimation to ensure that the estimates derived from the sample apply to the parameters of the entire population under study (e.g., every pixel in the mapped extent). Moreover, Stehman [11] highlights the impact of sample size allocation when using stratified random sampling for accuracy assessment and area estimation in the context of land-cover change mapping, with a focus on addressing the competing estimation objectives for rare and common classes, a frequent issue in the context of class imbalance.

A confusion matrix is a valuable tool for quantifying the performance of a classification algorithm [1,2]. An example confusion matrix is provided in Table 1. By comparing the independently determined labels, which define the columns of the table, and predicted classes, which define the rows of the table, of data points or sampling units not included in the training process (e.g., validation or testing data), it presents a comprehensive picture of model performance. This matrix goes beyond merely summarizing the overall accuracy or error, as it helps to quantify the specific types of errors. Analyzing the confusion matrix enables users to discern which classes are accurately predicted, which are inaccurately predicted, and which tend to be confused with each other [1,2]. Below, we discuss the impact of class imbalance on the confusion matrix and derived metrics, and highlight the utility of a population confusion matrix in which the relative proportion of samples approximates those within the landscape being mapped.

**Table 1.** Confusion matrix conceptualization where three classes, A, B, and C, are differentiated. *Pij* represents the proportion of samples classified as class *i*, but known to belong to class *j*. The + symbol is used to represent summation; when the + symbol occurs in the first subscript position, the rows are summed; when the + symbol is in the second subscript position, the columns are summed. Gray cells represent correct classifications. UA = user's accuracy (1 − commission error) and PA = producer's accuracy (1 − omission error).

| | | Reference | | | | |
| --- | --- | --- | --- | --- | --- | --- |
| | | **A** | **B** | **C** | **Row Total** | **UA** |
| | A | $P_{AA}$ | $P_{AB}$ | $P_{AC}$ | $P_{A+}$ | $P_{AA}/P_{A+}$ |
| Classification | B | $P_{BA}$ | $P_{BB}$ | $P_{BC}$ | $P_{B+}$ | $P_{BB}/P_{B+}$ |
| | C | $P_{CA}$ | $P_{CB}$ | $P_{CC}$ | $P_{C+}$ | $P_{CC}/P_{C+}$ |
| | Column total | $P_{+A}$ | $P_{+B}$ | $P_{+C}$ | | |
| | PA | $P_{AA}/P_{+A}$ | $P_{BB}/P_{+B}$ | $P_{CC}/P_{+C}$ | | |

OA is a commonly employed metric in traditional remote sensing accuracy assessment for evaluating the performance of classification models. It is calculated as the proportion of correctly classified testing samples among the total number of withheld samples [1,12]. Using the symbology from Table 1, OA = $(P_{AA} + P_{BB} + P_{CC})/P_{++}$, where $P_{++}$ represents

summation over both the rows and columns, and thus, $P_{++} = 1.0$. Alongside OA, it is common to calculate the class-level assessment metrics, PA and UA (see Section 1, above). PA represents a 1 – omission error, while UA represents a 1 – commission error [1].

The Kappa statistic has traditionally been calculated alongside OA as a measure of chance-adjusted agreement [13]. However, following decades of research highlighting its limitations, including the fact that it does not assume the reference labels are necessarily correct, the use of this metric is no longer regarded as useful in remote sensing accuracy assessment [14,15]. Therefore, Kappa will not be discussed further here.

In binary classifications, when only two classes are differentiated, it is common to label the class of interest as the positive case, and the background as the negative case, as described in Table 2. True positives (TPs) and true negatives (TNs) are, respectively, positive and negative case samples that are correctly mapped. On the other hand, false positives (FPs) and false negatives (FNs) are samples that are incorrectly mapped to the positive and negative classes, respectively (see Table 2).

**Table 2.** Conceptualization of binary confusion matrix and associated terminology. TP = true positive, TN = true negative, FN = false negative, FP = false positive, and NPV = negative predictive value. See Table 3 for equations for recall, precision, specificity, and NPV.

| | | Reference Data | | | |
|---|---|---|---|---|---|
| | | **Positive** | **Negative** | | **1 – Commission Error** |
| Classification Result | Positive | TP | FP | $\frac{TP}{TP + FP}$ | Precision |
| | Negative | FN | TN | $\frac{TN}{FN + TN}$ | NPV |
| | | $\frac{TP}{TP + FN}$ | $\frac{TN}{FP + TN}$ | | |
| | 1 – omission error | Recall | Specificity | | |

**Table 3.** Multiclass and binary metrics commonly calculated from the confusion matrix. TP = true positive, TN = true negative, FN = false negative, FP = false positive.

| Type of Classification | Metric | Equation | Comments |
|---|---|---|---|
| Binary and multiclass | Overall accuracy (OA) | $\frac{\text{Count of correct samples}}{\text{Count of total samples}}$ or $\frac{TP + TN}{TP + TN + FP + FN}$ | |
| Multiclass | User's accuracy (UA) | $\frac{\text{Count of correctly labeled samples in class}}{\text{Total count of samples predicted to class}}$ | 1 – commission error |
| | Producer's accuracy (PA) | $\frac{\text{Count of correctly labeled samples in class}}{\text{Total count of samples actually in class}}$ | 1 – omission error |
| Binary | Recall | $\frac{TP}{TP + FN}$ | PA for positives (1 – positive case omission error) |
| | Precision | $\frac{TP}{TP + FP}$ | UA for positives (1 – positive case commission error) |
| | Specificity | $\frac{TN}{TN + FP}$ | PA for negatives (1 – negative case omission error) |
| | Negative predictive value (NPV) | $\frac{TN}{TN + FN}$ | UA for negatives (1 – negative case commission error) |
| | F1-score (Dice score) | $\frac{2 \times \text{Precision} \times \text{Recall}}{\text{Precision} + \text{Recall}}$ or $\frac{2 \times TP}{2 \times TP + FN + FP}$ | |

Each of the UAs and PAs of the positive and negative classes are typically given their own names (Tables 2 and 3). Precision and negative predictive value (NPV) are equivalent to UA for positive and negative cases, respectively, while recall (also sometimes known as sensitivity) and specificity are, respectively, equivalent to PAs. As documented by Maxwell et al. [6], there is some confusion in the names used for class accuracies based on the binary model, and thus, it is important to always define the meaning of accuracy measures used in a study. The F1-score is commonly employed as a single metric that combines precision and recall, and is calculated as the harmonic mean of precision and recall. The F1-score considers both errors of omission, as estimated with recall, and errors of commission, as estimated with precision, relative to the positive case [12]. Table 3 summarizes these metrics, how they are calculated, and the relationships between them.

## 2.2. Averaged Multiclass Accuracy Measures

The binary metrics discussed above, including precision, recall, and F1-score, and the related loss metrics, which will be discussed in the next section, have been adapted for use in multiclass classification problems. However, varying approaches to aggregating the metrics across classes are used. For example, macro-averaged multiclass recall (Equation (1)) entails separately calculating recall for each class (*j*) across all classes (*C*), then summing the metrics, and finally, dividing by the number of classes (*N*) to obtain an average recall. Since all classes are averaged, each takes on equal weight in the calculation. Macro-averaged multiclass precision is an equivalent measure, in which the individual precision values for each class are combined in a simple arithmetic average (Equation (2)). Similarly, consistent with the basic F1 statistic definition (Table 3), the macro-averaged F1 statistic is simply the harmonic mean of the macro-averaged recall and macro-averaged precision.

$$\text{Macro-averaged multiclass recall} = \frac{1}{N}\sum\nolimits_{j=1}^{C} \frac{\text{TP}_j}{\text{TP}_j + \text{FN}_j} \tag{1}$$

$$\text{Macro-averaged multiclass precision} = \frac{1}{N}\sum\nolimits_{j=1}^{C} \frac{\text{TP}_j}{\text{TP}_j + \text{FP}_j} \tag{2}$$

Unfortunately, there is considerable confusion in the literature regarding terminology used for macro-averaging. Macro-averaged recall, macro-averaged precision, and macro-averaged F1 are also frequently referred to as, respectively, mean recall [16] (and sometimes, even more confusingly, labeled as mean accuracy [17]), mean precision [16], and mean F1 [18].

As an alternative to the equal weighting of classes, micro-averaged multiclass recall, precision, and F1 have been proposed as prevalence-dependent measures. In micro-averaged recall (Equation (3)), the total count of TPs across all *C* classes is summed and divided by the total count of all TPs and FNs across the *C* classes. The numerator in Equation (3) is therefore the sum of the correct samples, and the denominator is the overall total of the sum of each column, i.e., the sum of the entire confusion matrix. Thus, micro-averaged recall is equivalent to the definition of the OA statistic [8]. The associated micro-averaged precision statistic (Equation (4)) can also be shown to be equivalent to OA, and thus, micro-averaged recall and micro-averaged precision are identical. Furthermore, since the harmonic mean of identical values is itself equal to those values, micro-averaged recall, micro-averaged precision, and micro-averaged F1 are all identical and equal to OA [8].

$$\text{Micro-averaged multiclass recall} = \frac{\sum_{j=1}^{C} \text{TP}_j}{\sum_{j=1}^{C} \text{TP}_j + \sum_{j=1}^{C} \text{FN}_j} \tag{3}$$

$$\text{Micro-averaged multiclass precision} = \frac{\sum_{j=1}^{C} \text{TP}_j}{\sum_{j=1}^{C} \text{TP}_j + \sum_{j=1}^{C} \text{FP}_j} \tag{4}$$

It is also possible to calculate a weighted macro-averaged recall, precision, and F1-score where the contribution of each class in the final average is controlled by a user-defined weight ($w_j$) (Equations (5) and (6)) [7,8]. This allows the user to specify the relative weighting of each class in the aggregated metric, as opposed to traditional macro-averaging, where all classes are equally weighted, or micro-averaging, in which the values in the confusion matrix are aggregated prior to calculating the accuracy metric, and thus, the relative proportion of each class in the testing or validation dataset is preserved [3,7,8]. An important caveat is that when the class prevalences are used as the weights, weighted macro-averaged class metrics are also equivalent to OA.

$$\text{Weighted macro-averaged multiclass recall} = \frac{1}{\sum_{j=1}^{C} w_j} \sum_{j=1}^{C} \left( w_j \frac{\text{TP}_j}{\text{TP}_j + \text{FN}_j} \right) \quad (5)$$

$$\text{Weighted macro-averaged multiclass precision} = \frac{1}{\sum_{j=1}^{C} w_j} \sum_{j=1}^{C} \left( w_j \frac{\text{TP}_j}{\text{TP}_j + \text{FP}_j} \right) \quad (6)$$

### 2.3. Loss Metrics

When using backpropagation and mini-batch stochastic gradient descent (SGD) and its derivatives to iteratively update trainable parameters in CNN-based deep learning models, the loss metric serves as the sole measure of error to guide the learning process. As a result, the choice of an appropriate loss metric is of great importance [3,4,19,20]. The level of class imbalance potentially has a large impact on the suitability of a classification loss metric [3], as will be discussed below.

Binary cross-entropy (BCE) loss is the predominant loss metric used in binary classification tasks, while cross-entropy (CE) loss is common for multiclass classification tasks. The equations for these loss metrics are provided in Table 4. BCE and CE are examples of distribution-based-loss measures [4]. BCE loss is minimized when all $n$ case samples, coded as $y_i$, with values of 1 when positive and 0 when negative, are predicted to have a positive class probability ($\hat{p}_i$) approaching 1 for the former and 0 for the latter. CE loss also makes use of predicted class probabilities ($p_{ij}$) and is minimized when the class probability for each sample ($i$) within each class ($j$) approaches 1 for the correct class and 0 for all other classes. Classes with more samples will have a larger weight or impact in the calculation; as a result, CE loss is sensitive to class imbalance. In response, a weighted cross-entropy loss metric is sometimes used to specify the relative weights, $w_j$, of each class in the calculation. The values of $w_j$ are commonly based on the inverse of the abundance of the class in the training dataset in order to attempt to offset the impact of sample size. It is possible to increase the impact of difficult to classify samples by including a $\gamma$ parameter. This is known as focal loss [3].

**Table 4.** Commonly used loss metrics for binary and multiclass classification.

| Loss | Equation |
|------|----------|
| Binary cross-entropy (BCE) loss | $-\frac{1}{n} \sum_{i=1}^{n} [y_i \cdot \log(\hat{p}_i) + (1 - y_i) \cdot \log(1 - \hat{p}_i)]$ |
| Cross-entropy (CE) loss | $-\frac{1}{n} \sum_{i=1}^{n} \sum_{j=1}^{C} y_{ij} \cdot \log\left(\hat{p_{ij}}\right)$ |
| Weighted CE loss | $-\frac{1}{n} \sum_{i=1}^{n} \sum_{j=1}^{C} w_j \cdot y_{ij} \cdot \log\left(\hat{p_{ij}}\right)$ |

There are alternatives to BCE and CE loss and their derivatives. For example, there are several loss metrics that are derived from the Dice metric (an alternative name for the F1-score; see Table 3), and which are all region-based losses [4]. Since Dice is a measure of accuracy, the value of (1 – Dice) is used as the loss (i.e., error) metric [21–23]. Furthermore,

in order to make the loss metric differentiable, class probabilities are generally used, as opposed to hard labels. Equations (7) and (8) present a generalization of Dice loss for multiclass classification. Equation (7) represents a micro-averaged version, while Equation (8) provides a macro-averaged version. As with the F1-score, these loss metrics consider omission and commission errors relative to the positive case.

For Dice loss calculated using micro-averaging (Equation (7)), the predicted class probabilities relative to the correct class for the TPs ($\hat{p}_{TP}$) are summed and multiplied by two, and then, divided by the sum of $\hat{p}_{TP}$ multiplied by two, FN class probabilities relative to the predicted class ($\hat{p}_{FN}$), and FP class probabilities relative to the predicted class ($\hat{p}_{FP}$). A smoothing factor ($\varepsilon$) is commonly added to both the numerator and the denominator for computational stability and to exclude divide-by-zero errors. The result is then subtracted from 1 to convert from a measure of accuracy to an error metric. In Section 2.2, we explained that micro-averaging is equivalent to OA. The same logic applies to micro-averaged Dice loss, which is related to 1 – OA; however, it makes use of predicted class probabilities as opposed to predicted class labels, so is not strictly equivalent to OA.

In contrast to micro-averaging, the macro-averaged version (Equation (8)) calculates Dice loss separately for each class (*j*) of the C classes, and then, divides by the number of classes (*N*) to obtain an averaged Dice loss in which each class is equally weighted. It is also possible to calculate the weighted micro-averaged Dice loss where the user defines the relative weight of each class in the overall average [7,8,21,22,24,25].

$$\text{Micro-averaged Dice loss} = 1 - \left( \frac{(2 \times \Sigma\hat{p}_{TP}) + \varepsilon}{(2 \times \Sigma\hat{p}_{TP}) + \Sigma\hat{p}_{FN} + \Sigma\hat{p}_{FP} + \varepsilon} \right) \tag{7}$$

$$\text{Macro-averaged Dice loss} = \frac{1}{N}\sum_{j=1}^{C} \left( 1 - \left( \frac{(2 \times \Sigma\hat{p}_{TP}) + \varepsilon}{(2 \times \Sigma\hat{p}_{TP}) + \Sigma\hat{p}_{FN} + \Sigma\hat{p}_{FP} + \varepsilon} \right) \right) \tag{8}$$

Similar to CE loss, a focal version of Dice loss can be calculated by adding a $\gamma$ term to control the relative impact of difficult-to-classify samples [25]. Tversky loss is a modification of Dice loss that adds $\alpha$ and $\beta$ terms to control the relative weights of false positive and false negative errors. These loss measures were developed in response to the concern that highly imbalanced training data tend to result in a classification biased towards high precision and low recall for rare classes [26]. Therefore, by setting $\beta > \alpha$ (typical values are 0.7 and 0.3), FN is given increased weight in the loss function, and the resulting classification typically has an increased recall for rare classes, though this is likely at some cost to precision [26,27].

## 3. Methods

### 3.1. Data

To illustrate the issues involved in selecting accuracy and loss metrics, we used the EuroSat dataset [28] (Figure 1), which is a large and diverse dataset of satellite images that can be used to train and evaluate land use and land cover classification models. These data were generated for use in scene labeling or scene classification problems, where the entire image extent is labeled as a single class, in contrast to semantic segmentation, where each individual pixel is labeled. The dataset consists of 27,000 64-by-64-pixel (representing 640 by 640 m) image chips, each of which is labeled as one of ten land cover classes: "annual crop", "forest", "herbaceous vegetation", "highway", "industrial", "pasture", "permanent crop", "residential", "river", or "sea/lake" [28]. The images were captured over European countries by the Sentinel-2 satellite, which is operated by the European Space Agency (ESA), using the Multispectral Instrument (MSI) sensor. Of the 13 Sentinel-2 spectral bands (Table 5), three bands (B1, B9, and B10) are designed for atmospheric correction or quality control, and therefore, were excluded from our classification experiments. This left 10 bands, all of which were used as the input variables [29].

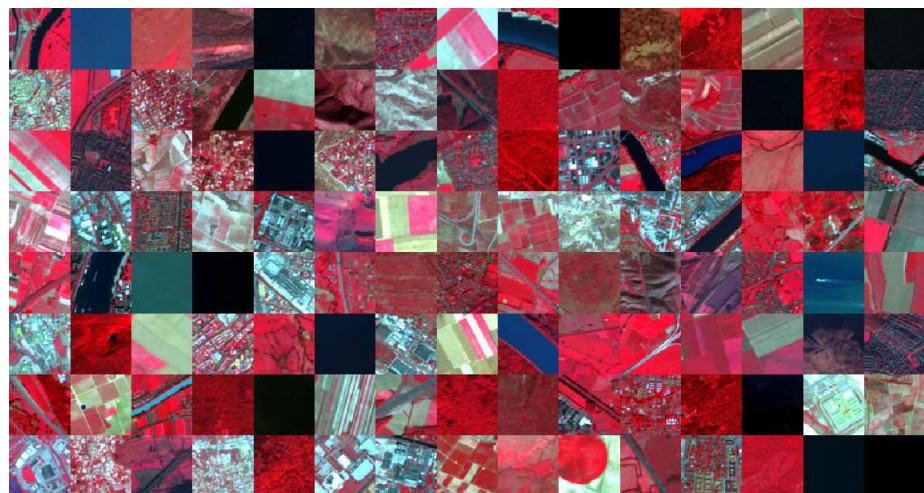

**Figure 1.** Example image chips from EuroSat dataset. Chips are displayed as false color composites, with the NIR, red, and green bands displayed, respectively, as red, green, and blue. Each chip is 64 by 64 pixels in size with a spatial resolution of 10 m.

**Table 5.** Sentinel-2 (MSI) bands. RE = red edge, NIR = near infrared, SWIR = shortwave infrared. Wavelength units are in nanometers (nm). Y indicates bands used in the experiment.

| Band | B1 | B2 | B3 | B4 | B5 | B6 | B7 | B8 | B8a | B9 | B10 | B11 | B12 |
|---|---|---|---|---|---|---|---|---|---|---|---|---|---|
| Central wavelength (nm) | 443 | 490 | 560 | 665 | 705 | 740 | 783 | 842 | 865 | 940 | 1375 | 1610 | 2190 |
| Description | Ultra blue | Blue | Green | Red | RE1 | RE2 | RE3 | NIR | NIR (narrow) | Water vapor | Cirrus cloud | SWIR 1 | SWIR 2 |
| Used in our experiment | | Y | Y | Y | Y | Y | Y | Y | Y | | | Y | Y |

The non-overlapping training, validation, and testing partitions defined by the data originators were used in this study. In order to generate a balanced training set, we used stratified random sampling without replacement to sample 1400 images from each class available in the original dataset, for a total of 14,000 samples (Table 6). With ten classes, the balanced dataset has a class prevalence of 10% for every class. We also generated an imbalanced training set, with half of the classes ("annual crop", "herbaceous vegetation", "industrial", "residential", and "sea/lake") randomly selected and assigned a reduced number of samples, 140, for an overall total of 7700 samples. In the imbalanced dataset, the class prevalences are 18.18% for each of the five common classes, and 1.82% for each of the five rare classes. This process was replicated for the validation datasets. We selected 400 samples from each class, or 4000 samples in total, for the balanced set. For the imbalanced validation dataset, we randomly selected a subset of 40 samples from the same five classes that were subsampled in the training set, resulting in an imbalanced validation set with 2200 samples. The test set used the original dataset class proportions and was not subsampled. Instead, after all samples were predicted and a sample error matrix was generated, we adjusted the proportions in the sample error matrix to represent population confusion matrices. Two separate test set population confusion matrices were generated: a balanced confusion matrix with equal prevalences for all the classes, and an imbalanced confusion matrix with class prevalence equivalent to the imbalanced training and validation sets.

**Table 6.** Number of samples in each class for the balanced and imbalanced training and validation sets. The testing set proportions used the samples in the original dataset, and the results were used to estimate balanced and imbalanced error matrices.

| | Training | | Validation | | Testing |
|---|---|---|---|---|---|
| | **Balanced** | **Imbalanced** | **Balanced** | **Imbalanced** | **Both** |
| Annual crop | 1400 | 140 | 400 | 40 | 300 |
| Forest | 1400 | 1400 | 400 | 400 | 300 |
| Herb veg | 1400 | 140 | 400 | 40 | 300 |
| Highway | 1400 | 1400 | 400 | 400 | 250 |
| Industrial | 1400 | 140 | 400 | 40 | 250 |
| Pasture | 1400 | 1400 | 400 | 400 | 200 |
| Perm crop | 1400 | 1400 | 400 | 400 | 250 |
| Residential | 1400 | 140 | 400 | 40 | 300 |
| River | 1400 | 1400 | 400 | 400 | 250 |
| Sea/Lake | 1400 | 140 | 400 | 40 | 359 |
| Total | 14,000 | 7700 | 4000 | 2200 | 2509 |

### 3.2. Classification Experiments

### 3.2.1. CNN Scene Classification

A series of experiments were performed using CNNs with the research objective of exploring the consequences of different averaging techniques within the context of scene classification using datasets with balanced and imbalanced classes. CNNs were defined and trained using the PyTorch library [30] in the Python language [31]. We designed the model to incorporate CNN design features that have been extensively used in the literature, rather than implementing state-of-the-art architectures that may be more rarely used, in order to ensure that our findings are relevant for typical CNN implementations. The CNN used a series of four 2D convolution, 2D batch normalization [32,33], rectified linear unit (ReLU) activation, and 2D max pooling layers followed by a series of two fully connected, 1D batch normalization, and ReLU activation layers. The final layer consisted of a fully connected layer that returned a logit for each of the ten classes. All 2D convolution layers used a kernel size of $3 \times 3$ with a stride of 1 and padding to retain the original array sizes in the spatial dimensions. The 2D max pooling used a kernel size of $2 \times 2$ and a stride of 2 to decrease the size of the array in the spatial dimensions by half. For the 2D convolution layers, 10, 20, 30, and 40 feature maps were generated, respectively. For the first two fully connected layers, 256 and 512 nodes or neurons were produced, respectively. Data augmentations and batch normalization were implemented to counteract overfitting and facilitate optimal convergence during training. For the data augmentations, we applied random horizontal and vertical flips using the albumentations Python library [34].

Each image in the training set had a 30% probability of undergoing these transformations. This probability setting ensured that each image had a substantial chance of being flipped either horizontally or vertically, thereby introducing necessary variability into the dataset. Such variability is particularly crucial when addressing issues of class imbalance in the data.

Our experiments were conducted using an NVIDIA GeForce RTX 3070 Ti GPU with 16 GB of GDDR5 memory. This GPU was paired with an Intel Core i9 processor and 16 GB of DDR5 RAM. This hardware configuration, combining a robust GPU with a powerful CPU and ample RAM, facilitated smooth and rapid processing of our CNN models. It allowed us to expedite iterations through various model architectures and parameter adjustments, essential for the success of our experimental setup.

Baseline experiments were executed using a range of loss functions. We used CE loss and Dice loss employing both macro- and micro-averaging techniques on the balanced and imbalanced datasets. We also trained a model using the imbalanced dataset and weighted CE loss in which the class weights were defined based on the inverse of their abundance in the training set. We did not train a model using the balanced dataset and weighted CE loss, since, with our weighting scheme, this would produce equal weights for the classes,

and thus, would be equivalent to using non-weighted CE loss. At the end of each training epoch, both the balanced and imbalanced validation data were predicted.

The training loop was executed for a maximum of 50 epochs using an AdamW optimizer with a default learning rate of 0.001. Instead of using the model state after 50 epochs, we selected the model that provided the best performance with the validation data, as measured using the validation loss. When using a balanced training set, the epoch that provided the best performance for the balanced validation dataset was selected. In contrast, and when using an imbalanced training set, the best epoch was selected based on the imbalanced validation dataset loss. During the training process, we used a large training and validation mini-batch size of 700 samples to ensure that minority class samples were included in each mini-batch. This required using multiple GPUs to train the model.

Accuracy assessment metrics were calculated from the prediction results obtained for the testing data. We used the R language and data science environment, and the yardstick [35], caret [36], rfUtilities [37], and differ [38] packages. We calculated the PAs (i.e., recalls) and UAs (i.e., precisions) for each of the ten classes. We also calculated OA and the class-aggregated recall, precision, and F1-score using both the macro- and micro-averaging methods.

### 3.2.2. Experiments Exploring the Effect of Changing Class Prevalences

We undertook an experiment with the research objective of exploring the impact of changing the relative proportions of each class in the testing set on the resulting assessment metrics. To accomplish this task, we used the sample2pop function provided in the differ R package [38]. The intended use of this function is to adjust the class proportions in a confusion matrix generated using stratified random sampling, where the strata are derived from the map itself. The output of the function is an estimate of the population confusion matrix, with row totals proportional to the area of each mapped class. In our case, however, we assumed the actual class prevalences were known, and thus, we wished to have instead column totals proportional to the class prevalences. We therefore transposed our confusion matrix prior to running the program, and afterwards, transposed the results back again. We used this function to generate 1000 random adjustments of the confusion matrix presented in Table 7 below with varying class proportions. This was accomplished by randomly selecting ten values that sum to 1.0 from a uniform distribution and adjusting the error matrix relative to these specified class proportions. From each generated confusion matrix, we then calculated all class PAs and UAs; the OA; and the class-aggregated PA (i.e., class-aggregated recall), UA (i.e., class-aggregated precision), and F1-score using both the macro- and micro- averaging methods.

**Table 7.** Example confusion matrix for classification of an imbalanced dataset using an imbalanced training set and cross-entropy loss. The table is an estimate of the population confusion matrix, and thus, the numbers in the table represent the percentage of the imbalanced data classified as class *i*, but known to belong to class *j*.

| | | Reference | | | | | | | | | | | |
| | | Annual Crop | Forest | Herb Veg | Highway | Industrial | Pasture | Perm Crop | Residential | River | Sea/ Lake | Row Total | UA |
|---|---|---|---|---|---|---|---|---|---|---|---|---|---|
| Classification | Annual crop | 15.45 | 0.00 | 0.30 | 0.00 | 0.00 | 0.00 | 0.04 | 0.00 | 0.00 | 0.00 | 15.79 | 0.979 |
| | Forest | 0.24 | 1.79 | 0.12 | 0.00 | 0.00 | 0.02 | 0.00 | 0.00 | 0.00 | 0.00 | 2.18 | 0.825 |
| | Herb veg | 0.06 | 0.00 | 13.09 | 0.01 | 0.07 | 0.03 | 0.01 | 0.06 | 0.00 | 0.00 | 13.33 | 0.982 |
| | Highway | 0.30 | 0.00 | 0.91 | 1.75 | 4.73 | 0.01 | 0.01 | 1.58 | 0.04 | 0.00 | 9.33 | 0.188 |
| | Industrial | 0.00 | 0.00 | 0.00 | 0.00 | 12.51 | 0.00 | 0.00 | 0.24 | 0.00 | 0.00 | 12.75 | 0.981 |
| | Pasture | 0.61 | 0.02 | 0.85 | 0.00 | 0.00 | 1.76 | 0.01 | 0.00 | 0.01 | 0.00 | 3.27 | 0.539 |
| | Perm crop | 1.52 | 0.00 | 2.67 | 0.03 | 0.00 | 0.00 | 1.72 | 0.30 | 0.00 | 0.00 | 6.24 | 0.276 |
| | Residential | 0.00 | 0.00 | 0.00 | 0.01 | 0.65 | 0.00 | 0.01 | 16.00 | 0.00 | 0.00 | 16.68 | 0.959 |
| | River | 0.00 | 0.00 | 0.24 | 0.01 | 0.22 | 0.00 | 0.01 | 0.00 | 1.77 | 0.00 | 2.81 | 0.630 |
| | Sea/lake | 0.00 | 0.00 | 0.00 | 0.00 | 0.00 | 0.00 | 0.00 | 0.00 | 0.00 | 17.62 | 17.62 | 1.000 |
| | Column total | 18.18 | 1.82 | 18.18 | 1.82 | 18.18 | 1.82 | 1.82 | 18.18 | 1.82 | 18.18 | | |
| | PA | 0.850 | 0.987 | 0.720 | 0.964 | 0.688 | 0.970 | 0.948 | 0.880 | 0.972 | 0.969 | | |

## 4. Results, Discussion, and Recommendations

### 4.1. Micro- and Macro-Averaged Accuracy Assessment Metrics

Table 7 provides the accuracy assessment results for the CNN-based scene classification model described above, trained using the imbalanced training set and standard CE loss, and applied to the withheld testing data, with the confusion matrix adjusted to the imbalanced class proportions. The numbers in the table represent the percentages of the assumed imbalanced population. The OA for this classification is 0.835, and the class PAs and UAs, included on the table margins, vary from 0.688 to 0.972 and from 0.188 to 1.000, respectively.

Table 8 provides the class-aggregated metrics calculated from Table 7. It is notable that the micro-averaged multiclass metrics, micro-UA (i.e., micro-precision), micro-PA (i.e., micro-recall), and micro-F1-score, are identical to the OA value. This is because, as was explained in Section 2.2 [8], micro-averaged multiclass metrics are directly equivalent to OA. The differentiation between FP and FN that is central to a binary approach to accuracy assessment (see Table 2) breaks down with the application of micro-averaging to multiclass accuracy measures. For example, in Table 7, 0.04% of the imbalanced test dataset comprises "permanent crop" reference samples incorrectly classified as "annual crop," and represent errors of omission or FNs from the perspective of the "permanent crop" class. However, this same 0.04% of samples represent commission errors (FPs) relative to the prediction of the "annual crop" class. Thus, every non-diagonal element is both an FN and an FP, and therefore, summing TPs and FNs (recall; Equation (3)), or TPs and FPs (precision; Equation (4)), results in the same overall total, equal to the sum of the matrix. Thus, micro-averaged multiclass recall and precision both represent the division of the sum of TPs by the confusion matrix total, i.e., OA. Similarly, since the F1-score is the harmonic mean of precision and recall, and the harmonic mean of two identical numbers is also identical to those two numbers, the micro-averaged F1-score is also equivalent to the micro-averaged precision and micro-averaged recall, as well as OA. Grandini et al. [8] provide further elaboration on the equivalency of the micro-averaged metrics and OA.

**Table 8.** Class-aggregated accuracy metrics for the classification reported in Table 8 (i.e., using an imbalanced training set and a cross-entropy loss).

| | Micro | | | Macro | | |
|---|---|---|---|---|---|---|
| OA | UA (Precision) | PA (Recall) | F1-Score | UA (Precision) | PA (Recall) | F1-Score |
| 0.835 | 0.835 | 0.835 | 0.835 | 0.736 | 0.895 | 0.755 |

Since OA is a well-known and intuitive statistic, with well-understood properties, it is not only unnecessary and redundant to use micro-averaged measures, but their use is likely to add confusion. Perhaps more importantly, since OA is not a useful measure for assessing class-level statistics, micro-averaged recall, micro-averaged precision, and micro-averaged F1 are also not useful for that purpose.

In contrast to micro-averaged metrics, macro-averaged metrics are unique and not equivalent to OA. Since each metric is calculated separately for each class, and then, subsequently averaged, this is not equivalent to simply dividing the number of TPs by the confusion matrix total.

### 4.2. Accuracy Assessment Metrics and Class Imbalance

Figure 2 summarizes the class-level UA and PA results for CE loss and a balanced training set, with 1000 replicates of varying class proportions, generated using the diffeR package's sample2pop function [38], which converts a sample confusion matrix to a population matrix. In calculating the population confusion matrices, class-level PAs are not impacted by changes in the relative class proportions in the reference dataset, since each class's PA is calculated only from samples in that particular reference class. In contrast, class-level UAs are affected by changing class proportions in the reference classes, since a

class's UA is calculated from the samples labeled as that class, and the number of samples from other classes incorrectly labeled as the class of interest is affected by the prevalence of those classes. For example, in Table 8, there is some confusion between the "annual crop" and "permanent crop" classes. If the relative proportion of "permanent crop" samples in the population increases, this will likely reduce the UA for the "annual crop" class, since there will be more opportunities for misclassification. In contrast, this will not affect the PA for the "annual crop" class, since this metric depends only on the samples belonging to the "annual crop" reference class.

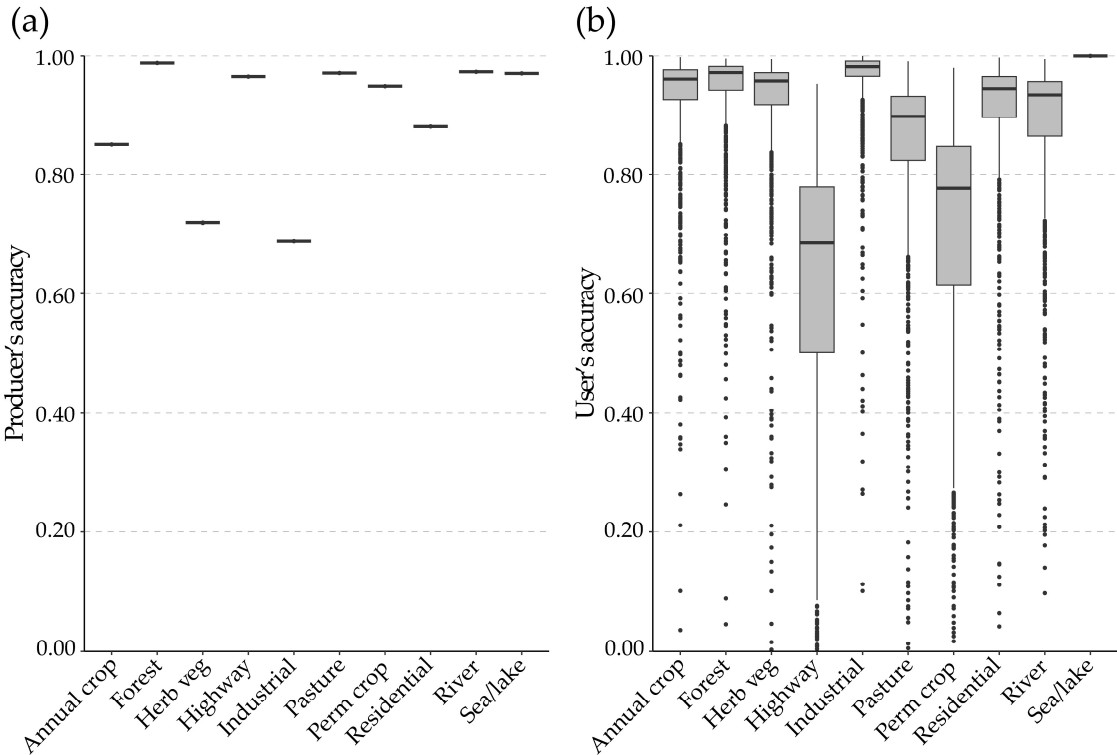

**Figure 2.** Boxplots of class PAs (**a**) and UAs (**b**) for 1000 randomly generated class proportions that sum to 1, selected from a uniform distribution.

It is also important to note that, as has been pointed out by Foody [14], although PA as a statistic is not prevalence-dependent, remote sensing classifiers themselves may be prevalence-dependent. Furthermore, even if classifiers were not prevalence-dependent, simply focusing on PAs and ignoring UAs would not provide a method for avoiding the problem of class prevalence, since PAs alone are not useful in characterizing class-based classification performance.

It is notable in Figure 2 that the range of UAs varies greatly by class. However, some classes, such as "forest" and "sea/lake", are consistently well mapped, with UAs close to 1.0. In contrast, the two classes with the lowest median UAs, "highway" and "permanent crop", have the largest interquartile range of UA values and the largest number of instances of UAs less than 0.1. This illustrates the point that although UAs are affected by class prevalence, the effect is smaller for classes with higher accuracies, tending to zero for classes mapped with a UA of 1.0. Furthermore, it is worth emphasizing that in Figure 2b, the UA for each class is generally lowest when that class's prevalence is lowest.

Figure 3 shows the results for OA and the class-aggregated, macro-averaged metrics. Since the micro-averaged metrics are equivalent to OA (see Section 4.1), they are not shown. OA varies with changes in class proportions. This is expected, since the relative proportions of samples will change the number of correct and misclassified samples in the table. For example, Table 8 shows that the "sea/lake" class was generally well differentiated from

the other classes. As a result, if a larger proportion of the testing samples were from the "sea/lake" class, higher OA accuracy and related metrics would be expected. The macro-averaged UA and F1-score are sensitive to class prevalence, while the macro-averaged PA is not. This is because the individual-class PAs are not sensitive to class prevalence, and therefore, their averages will also not be affected. On the other hand, since UAs are affected by class prevalence, their averages will also be affected, as will their F1-scores, since they rely in part on UAs.

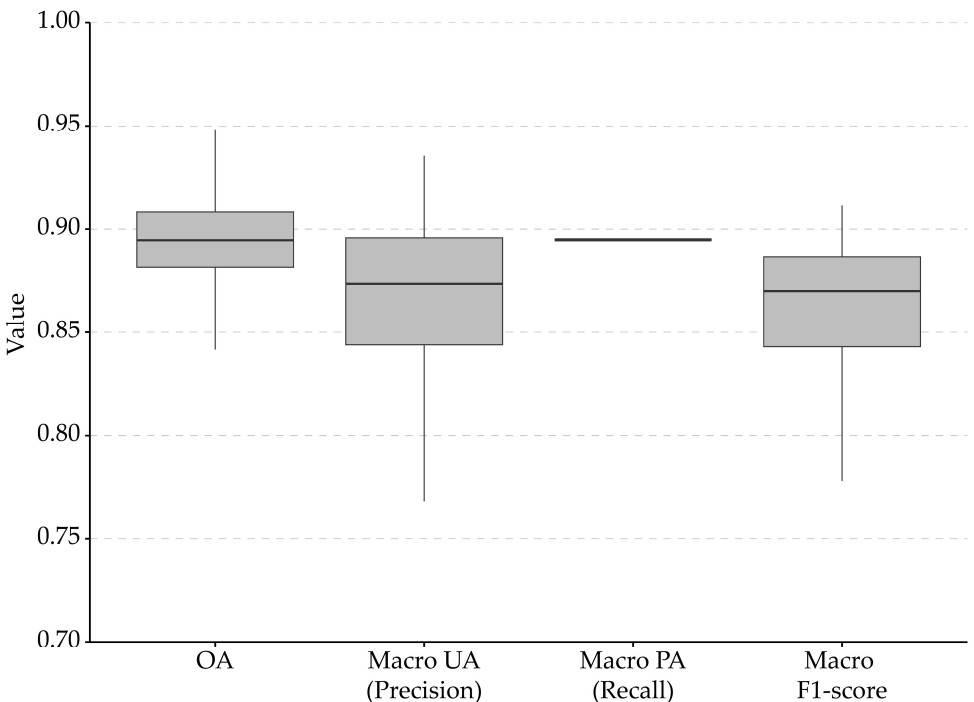

**Figure 3.** Boxplots of OA and class-aggregated, macro-averaged assessment metrics using 1000 varying class proportions that sum to 1, selected from a uniform distribution.

As noted above, accuracy assessment should take into account the relative class proportions in the landscape being mapped, since class prevalence has an effect on classification accuracy [2,5,10,39–43]. If simple random sampling is used to collect testing or validation samples, then the resulting confusion matrix will be an unbiased estimate of the population matrix, and therefore, can be used directly to estimate class accuracies. If some other probability-based sampling method is used, such as class-stratified random sampling, an approach which is often used when there are particularly rare classes, the resulting confusion matrix will have different class proportions from the thematic map being assessed [44]. In this circumstance, either the sample confusion matrix should be adjusted to reflect the population proportions, or the correction may be incorporated directly into the accuracy estimator formulas [5].

The importance of using reference sample proportions that align with landscape proportions for accuracy assessment provides a useful framework for selecting and interpreting class-aggregated assessment metrics. If the confusion matrix is based on a number of samples that are deliberately chosen to be uniform in each class, and the confusion matrix is not subsequently adjusted to represent population proportions, then the class-level UAs, class-level F1-scores, and OA will be representative of the accuracies of a hypothetical map with equal class prevalence. Since most real maps have at least one rare class, and rare classes are more difficult to map, the value of simulating a hypothetical equal-prevalence map is unclear. However, the class-level PAs will not vary with changes in class proportions [5].

Macro-averaging is based on a simple average after the statistics are calculated, and therefore, macro-averaged measures give equal weighting to each class's UA or PA that is

combined with the integrated metric. Some analysts favor macro-averaged class statistics precisely because rare classes are given equal weight to more common classes. However, UAs and, thus, F1 statistics are sensitive to class proportions, and rare classes are inherently more difficult to classify. Therefore, it is essential to use an estimate of the population matrix for the calculations to ensure that rare classes are not treated over-optimistically in the calculation of UAs and F1-scores.

Weighted macro-averaging methods are another way in which analysts sometimes attempt to deal with imbalance. Typically, class weights used for aggregating PAs are derived from the relative abundance of classes on the landscape, which can be estimated based on the column totals in a population confusion matrix set up, as in Table 1. Similarly, the class weights used for aggregating UAs are class prevalences in the classified map, and, in turn, can be estimated using the row totals in the population error matrix. However, when such an approach is used, these measures are equivalent to the micro-averaged PA and UA, respectively, and thus, in turn, are equivalent to OA. Prevalence-weighted macro-averaging is redundant, and therefore not a useful approach.

In summary, in calculating summary multi-class accuracy measures, macro-averaging is the only approach that appears to provide a useful average class-based measure, since micro-averaged measures are redundant with OA. Weighted macro-averaged measures that use class prevalence as the weights are also redundant. Neither micro-averaged measures nor prevalence-weighted macro-averaged measures should be reported, since the equivalent OA is a conceptually simpler term. Class prevalence affects the calculation of UA and F1-scores, and therefore, accuracy measures should always be estimated from the population confusion matrix. Although macro-averaged PA is not affected by class prevalence, it has been suggested that remote sensing classifiers may be affected by prevalence [14]. Since there is so much confusion in the literature, we recommend that authors clearly document their methods.

### 4.3. Impact of Class Imbalance on the Training Process

In Sections 4.1 and 4.2 above, we focused on assessing map accuracy. In this section, we explore the impact of class imbalance in the context of choosing a loss metric. On one level, determining the final map accuracy and loss metrics are very similar tasks, as both deal with assessing classification performance. However, an important distinction is that a key purpose of map accuracy assessment is usually communication of the uncertainty in the classification results. In contrast, losses calculated during training are primarily designed to guide the classifier towards an optimal model. Although not usually explicitly articulated, an optimal model is generally conceptualized as one in which most classes are classified well, which implies the need for a multiclass average of some sort. To explore these issues, we trained models using both a balanced and an imbalanced training set, and with the CE, macro-averaged Dice, and micro-averaged Dice losses. We also trained a model using weighted CE, where the class weights were defined based on the inverse of their abundance in the imbalanced training dataset. Since using weighted CE loss and our weighting scheme with a balanced dataset would be equivalent to unweighted CE with a balanced dataset, this combination was not tested; as a result, a total of seven models were trained.

Figure 4 shows the loss for the training, balanced validation set, and imbalanced validation set across the 50 training epochs, while Figure 5 shows the overall accuracy, and Figure 6 shows the macro-averaged F1-score. These results generally suggest that the choice of loss metric is less important when a balanced training set is used, as the CE and dice losses provided similar performance. However, the choice of loss function had a larger impact when the training set was imbalanced. For the imbalanced training data, loss metrics indicated a slower improvement between successive epochs compared to the balanced data training, and there was a notably larger divergence between the training loss metric and the associated validation loss. Furthermore, models trained using CE as the loss metric generally required a greater number of epochs to stabilize compared to using the

alternative Dice loss metrics, although after 50 epochs, the metrics indicated comparatively high accuracy for the CE training.

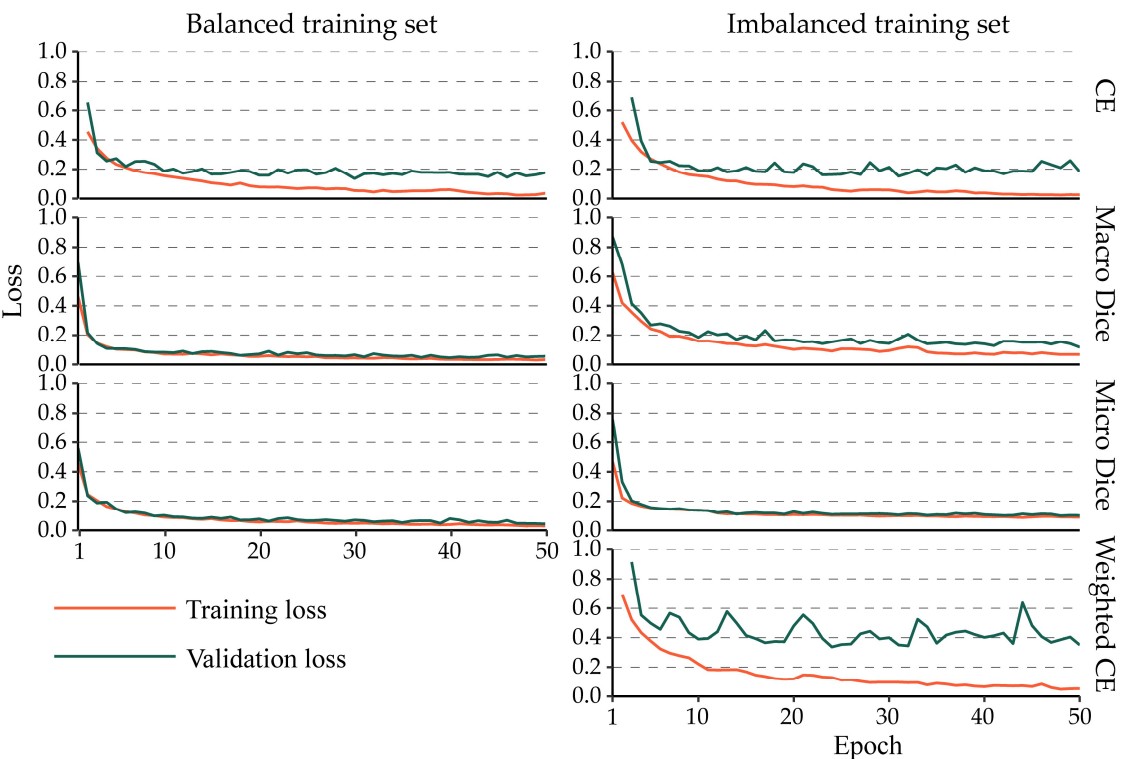

**Figure 4.** Loss curves for training of EuroSat dataset for 50 epochs using four different losses and balanced and imbalanced training sets.

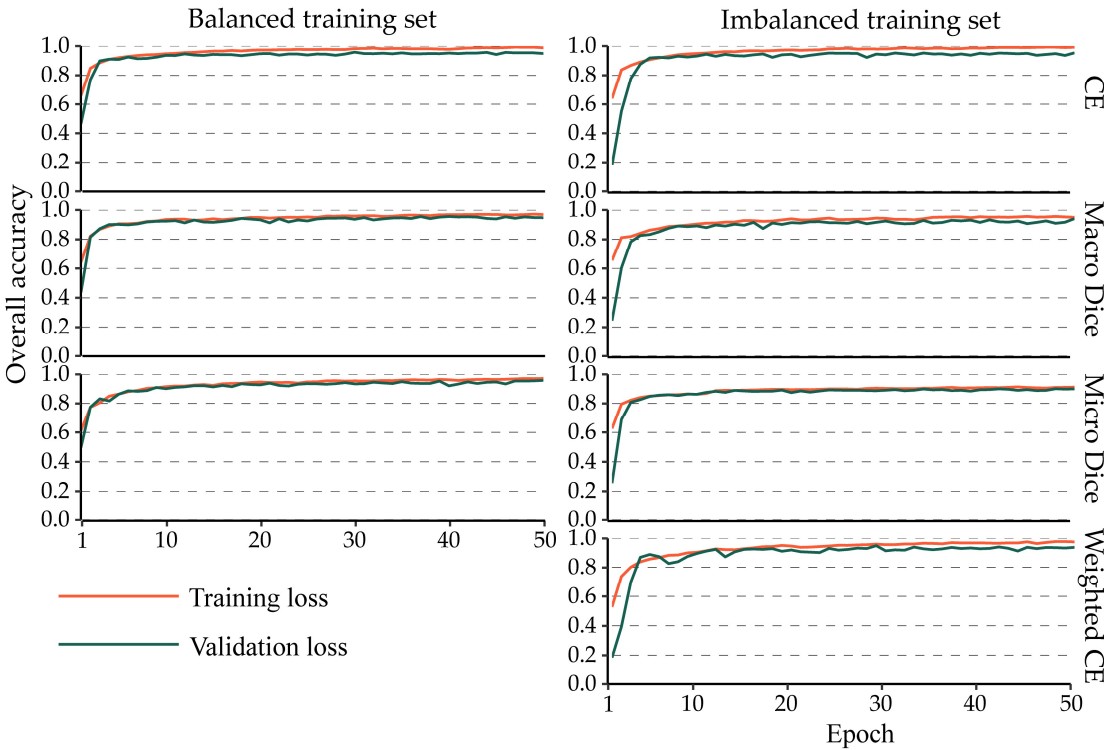

**Figure 5.** Overall accuracy for training of EuroSat dataset for 50 epochs using four different losses and balanced and imbalanced training sets.

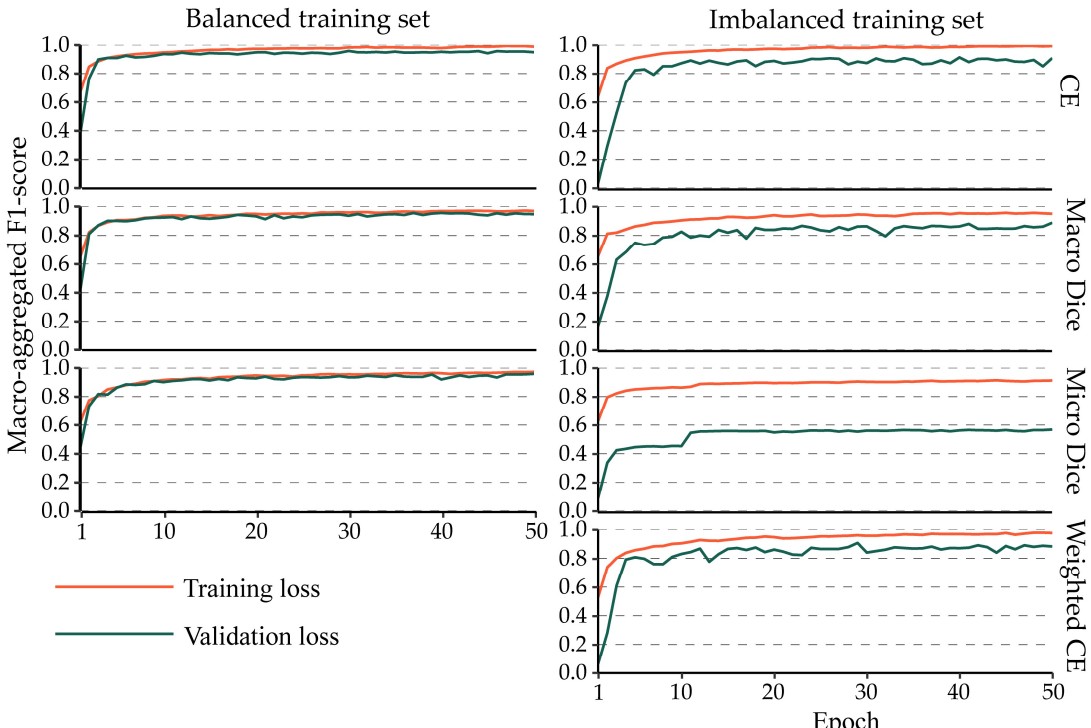

**Figure 6.** Macro-averaged class-aggregated F1-score for training of EuroSat dataset for 50 epochs using four different losses and balanced and imbalanced training sets.

For models trained with balanced data, micro-Dice appears to be a potentially appropriate loss metric, resulting in models with high accuracy values, as indicated by the validation data. However, for the imbalanced data, the use of micro-Dice loss for training resulted in consistently low macro-F1 trends (Figure 6), despite the moderate-to-high overall accuracy values (Figure 5). These figures illustrate the problem with micro-Dice as a loss measure. Since micro-Dice is similar to OA, except that class probabilities are used in the calculation as opposed to class labels, it is particularly unsuitable for imbalanced data.

The aggregated assessment metrics obtained when using the resulting models to predict the withheld testing data are provided in Table 9. Generally, regardless of the loss metric used, training and testing using the balanced datasets yielded higher model accuracy than training and testing using imbalanced data. This was true for all accuracy measures, including OA and the class-averaged measures of macro-F1, macro-UA, and macro-PA. This is not surprising, since imbalanced datasets are inherently more difficult to train than balanced datasets. However, the difference between the accuracy measures of the balanced training and testing, compared to the imbalanced training and testing, was smaller for CE and macro-Dice compared to micro-Dice. In addition, micro-Dice performed particularly poorly with imbalanced training and testing, resulting in the lowest accuracy measures observed. This emphasizes that micro-Dice, since it is similar to OA, is not effective as a loss measure for training with imbalanced data.

When class prevalences differed between training and testing (e.g., balanced training and imbalanced testing, a common approach, or the less common imbalanced training and balanced testing), models trained with CE and weighted CE generally resulted in high accuracies. Macro-Dice loss produced the highest accuracies for balanced training and imbalanced testing. However, imbalanced training followed by testing with balanced data resulted in low accuracies for macro-Dice and very low accuracies for micro-Dice.

**Table 9.** Comparison of assessment metrics for models trained using different loss functions and with balanced or imbalanced training sets.

| Loss Metric | Training Set | Test Prevalences | OA | Macro-F1 | Macro-UA (Precision) | Macro-PA (Recall) |
|---|---|---|---|---|---|---|
| CE | Balanced | Balanced | 0.959 | 0.958 | 0.959 | 0.959 |
|  | Balanced | Imbalanced | 0.970 | 0.929 | 0.907 | 0.958 |
|  | Imbalanced | Balanced | 0.895 | 0.895 | 0.911 | 0.895 |
|  | Imbalanced | Imbalanced | 0.835 | 0.755 | 0.736 | 0.894 |
| Weighted CE | Imbalanced | Balanced | 0.912 | 0.912 | 0.916 | 0.912 |
|  | Imbalanced | Imbalanced | 0.890 | 0.804 | 0.765 | 0.911 |
| Micro-Dice (OA) | Balanced | Balanced | 0.954 | 0.954 | 0.955 | 0.954 |
|  | Balanced | Imbalanced | 0.956 | 0.913 | 0.883 | 0.954 |
|  | Imbalanced | Balanced | 0.581 | 0.466 | 0.452 | 0.581 |
|  | Imbalanced | Imbalanced | 0.261 | 0.264 | 0.251 | 0.581 |
| Macro-Dice | Balanced | Balanced | 0.956 | 0.956 | 0.957 | 0.956 |
|  | Balanced | Imbalanced | 0.971 | 0.927 | 0.904 | 0.956 |
|  | Imbalanced | Balanced | 0.873 | 0.872 | 0.894 | 0.873 |
|  | Imbalanced | Imbalanced | 0.805 | 0.712 | 0.697 | 0.872 |

To summarize Table 9, the optimum training strategy in our experiments was to use balanced training data, irrespective of whether the modeled population (and the prevalences in the testing data) was balanced or imbalanced. Balanced training data not only led to higher accuracies, but the differences in accuracies between the loss metrics were small, making the choice of loss metric less important. However, in many situations, it is not possible to build balanced training data, and thus, training has to be carried out with imbalanced training data. With imbalanced training, using weighted CE as the loss metric generally produced the highest accuracy. Conveniently, weighted CE for balanced training is equivalent to CE, and thus, weighted CE provides the best overall choice for a loss metric, whether the data are balanced or imbalanced.

Figure 7 shows the UAs and PAs calculated for the withheld testing data for each class when using each loss metric, and balanced and imbalanced training sets to train the model, and balanced and imbalanced test sets to assess the model. Of note is that for training with the micro-Dice loss, the UAs and PAs for the five rare classes were generally low, indicating that the model learned to ignore these less abundant classes since they had a small impact on the loss metric. These findings highlight the conclusion that micro-averaged Dice loss is inappropriate when classes are highly imbalanced, because micro-averaged Dice is equivalent to OA. The class imbalance had less of an impact when using CE, weighted CE, or macro-averaged Dice loss, although the UAs and PAs for the less abundant classes were nevertheless still lower than when the balanced training dataset was used. We attribute this in part to the reduced sample size for these classes, which could result in the complexity of these classes not being represented in the training set. In other words, the low sample size may be confounding the class imbalance issue.

In summary, our experiments suggest that choosing a loss metric may not be as complex as has been thought. Dice loss is often suggested as an alternative to CE loss when classes are imbalanced [3,23], and the fact that there are choices between the micro- and macro- averaging of Dice would seem to make choosing between them difficult. However, since micro-averaged Dice loss is similar to $1 - OA$ and is sensitive to class imbalance, it can be ruled out as a choice. Therefore, if an integrated Dice loss measure is used, this only leaves macro-averaged Dice loss, but macro-Dice only performed well with balanced training, suggesting that it is not particularly robust in the presence of class imbalance. Therefore, we conclude that CE loss with class weightings offers the best overall choice for a loss metric. When the training data classes were imbalanced, weighted CE resulted in models with the highest accuracy, and when the training data were balanced, weighted

CE was equivalent to CE without weighting, the loss metric that resulted in the highest accuracy for balanced training.

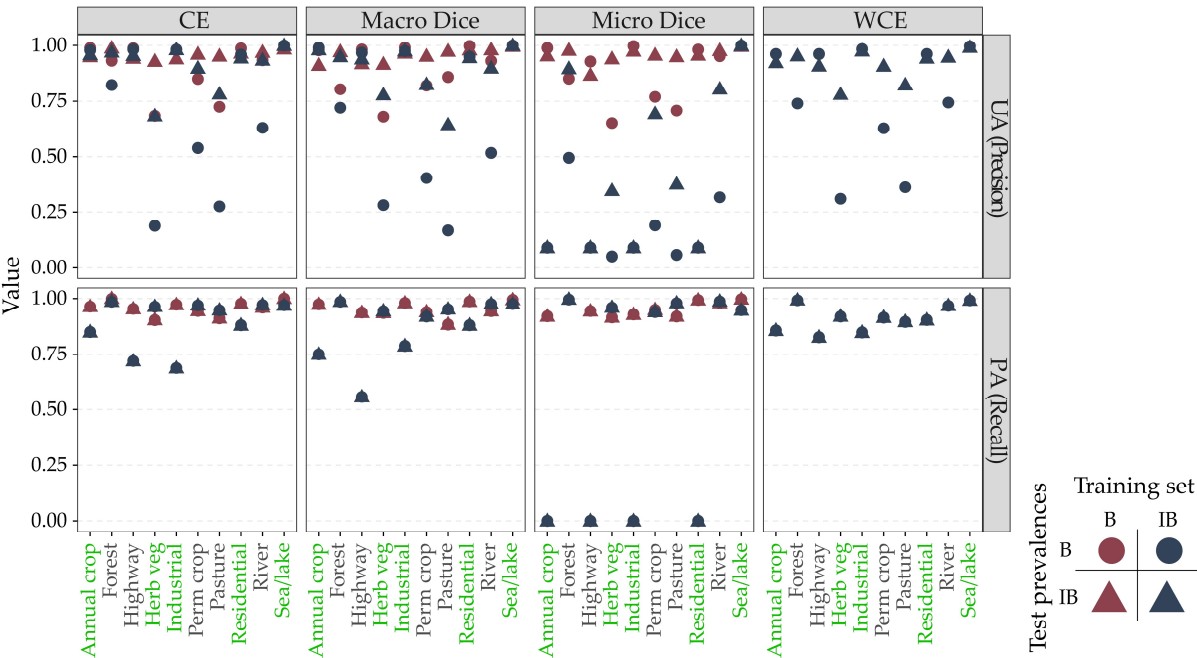

**Figure 7.** Class-level UA and PA metrics of CNN models trained using cross-entropy, weighted cross-entropy, and macro- and micro-averaged Dice loss metrics. Results using balanced and imbalanced training sets are differentiated by color. The names of rare classes, with a reduced number of samples in the training set, are printed in green.

Our results are also informative for other commonly used classification loss metrics that offer augmentations of CE or Dice loss. As noted above, Tversky loss offers an augmentation of Dice loss that allows the user to specify relative weightings for FP and FN errors using $\alpha$ and $\beta$ parameters, respectively [25–27]. Although this metric is commonly used for binary classification problems, especially when the positive class is rare, it can be modified for multiclass problems. Equation (9) provides a micro-averaged multiclass version of the metric, while Equation (10) provides a macro-averaged multiclass version. Since the micro-averaged version does not differentiate between classes, but instead, uses the total counts of TP, FN, and FP samples and associated probabilities to calculate the loss, different $\alpha$ and $\beta$ terms cannot be specified for each class. When micro-averaging is used, FP samples relative to one class become FN samples relative to other classes; as a result, the value and meaning of adding $\alpha$ and $\beta$ terms is unclear in the context of micro-averaging. However, when using a macro-averaged version, the user could specify different $\alpha$ and $\beta$ terms for each class to allow for differing class-level relative weightings of FP and FN errors. It is also possible to calculate the class-weighted macro-averaged Tversky loss. As a result, the user has control over relative class weightings in the final metric along with the relative weightings of FP and FN errors for each class separately. Although this allows for a high degree of refinement and customization, the tradeoff is increased complexity in configuring the metric. Based on our findings relating to Dice loss, we argue that a macro-averaged version of multiclass Tversky loss can be considered if classes are imbalanced; however, a micro-averaged version would not be appropriate. In regard to focal versions of the losses, such as focal CE loss [45] and focal Tversky loss [27], in which the primary goal is to control the relative impact of difficult-to-predict samples based on their prediction confidence, we argue that the results highlighted above for the associated non-focal versions hold for the augmented, focal versions. For example, if classes are imbalanced, a focal version of

weighted CE or macro-averaged multiclass Tversky loss is preferable to focal CE or focal micro-averaged multiclass Tversky loss.

$$\text{Micro-Averaged Multiclass Tversky Loss} = 1 - \left( \frac{\Sigma \hat{p}_{TP} + \varepsilon}{\Sigma \hat{p}_{TP} + \alpha \Sigma \hat{p}_{FP} + \beta \Sigma \hat{p}_{FN} + \varepsilon} \right) \quad (9)$$

$$\text{Macro-Averaged Multiclass Tversky Loss} = \frac{1}{N} \sum_{j=1}^{C} 1 - \left( \frac{\Sigma \hat{p}_{TP} + \varepsilon}{\Sigma \hat{p}_{TP} + \alpha \Sigma \hat{p}_{FP} + \beta \Sigma \hat{p}_{FN} + \varepsilon} \right) \quad (10)$$

It is important, however, to note that these findings may not hold for all problems or use cases. The performance of these different metrics may be case-specific, so the user may need to experiment with multiple loss metrics. Furthermore, different metrics may be appropriate for different stages of the analysis. For example, during the training phase, the primary concern is to produce an effective model. How to define its effectiveness is left up to the user, and questions of how the loss metrics relate to map accuracy may be of only minimal interest. At this initial stage, the emphasis may be on providing sufficient samples to train the model and ensure that all classes meet some basic minimum classification accuracy. On the other hand, during the final accuracy assessment stage, the metrics should have intuitive meaning, and most importantly, quantify real properties of the map itself, rather than of a hypothetical map. This requires that the accuracy be calculated from an estimate of the population confusion matrix.

It is also important to consider these findings within the broader context of research into class imbalance and classification. In their review paper, Ghosh et al. [46] point out that the class imbalance problem is tightly connected to dataset size issues and concept complexity (which, in turn, includes class separability). In our results, this is illustrated by the consistently high classification accuracies of the sea/lake class, irrespective of the loss metric used, despite its rarity in the imbalanced training data (Figure 7). The distinct spectral and spatial properties of water presumably compensate for any problems due to the limited number of training samples. On the other hand, the spectrally variable industrial and highway classes both suffer from large declines in PA (recall), when trained with the imbalanced data, despite the fact that, of the two classes, only industrial is a rare class in the imbalanced dataset (Figure 7).

Ghosh et al. [46] also identify three categories of approaches for dealing with class imbalance in the deep learning literature: those that are applied through pre-processing of the input data, post-processing of the predictions, and special purpose algorithms (see also Johnson and Khoshgoftaar [47]). For example, at the algorithmic level, Ding et al. [48] found that a very deep architecture, with greater than 10 layers in the CNN, increased accuracy for the classification of imbalanced classes. Addressing imbalance through the selection of an appropriate loss metric is also an algorithmic approach [46]. Loss metrics represents a particularly attractive way of addressing imbalance because it is a conceptually simple approach, directly affects the decision boundary, and places very little, if any, additional burden on the classifier.

## 5. Conclusions

When training and assessing classification models with imbalanced classes in the training set and/or within the landscape being mapped, it is important to consider the appropriate use and interpretation of accuracy assessment metrics. In this paper, we focused on choosing between and interpreting class-aggregated accuracy assessment metrics and multiclass loss metrics. Here, we conclude with best practices and recommendations.

All final summary accuracy assessment statistics, including both the statistics of individual classes and averages over the classes, should be calculated from a population confusion matrix. An important feature of the population matrix is that it incorporates the class prevalences, which are a fundamental feature of the classification. For example, if OA is calculated from a confusion matrix that does not take into account landscape proportions, or appropriate corrections are not made in the analysis, the reported OA will

be biased, since it is sensitive to relative class proportions. PA (recall) is usually listed as a metric insensitive to class prevalence, with the implication that it can be calculated from the sample confusion matrix. However, the classifier itself, rather than the statistic, may be prevalence-dependent [14]. Thus, all accuracy statistics, including OA, UA (precision), PA (recall), F1-score, and their macro-averages, should be calculated from population confusion matrices.

Macro-averaged assessment metrics provide a summary of how well classes are differentiated on average. When class-level UA and PA are aggregated using the micro-averaging method, they are equivalent to OA. Since these metrics are redundant, they should not be reported, as doing so is only likely to add confusion. Weighted macro-averaging, where the weights are the class prevalences, is also equal to OA, and thus, should also be avoided.

Loss metrics, though they are a measure of classification success, serve a purpose that differs from that of final map accuracy assessment. Loss metrics guide the model development, and thus, affect the accuracy of the final product. The communication of map uncertainty is generally not relevant for loss metrics, and thus, it is not necessary to calculate the loss from estimates of the population. Our examples demonstrate that, since users typically wish to maximize class accuracy across all classes, balanced training data are preferable to imbalanced data, irrespective of whether the actual dataset to be classified is imbalanced or not. In practice, however, generating a balanced training dataset may not be possible, and thus, consideration of how loss metrics are affected by class imbalance is important.

In our experiments, the CE loss metric with weighting proportional to class prevalences in the training data generally resulted in models with the highest accuracy statistics. This was true for models trained with both balanced training data (in which case, the loss metric is equivalent to CE without weighting) and also for imbalanced training data. This is an important finding, since as noted above, developing balanced training data is challenging for many applications. Dice loss has been suggested as an alternative to CE loss when classes are imbalanced. However, the type of averaging of Dice losses is important. Macro-Dice is a simple average of the individual F1-statistics, thus weighting all classes equally. Macro-Dice loss resulted in accuracies similar to, but consistently slightly lower than, the accuracies obtained with the CE loss without weighting. Micro-averaged dice loss is similar to $1 - OA$, except that class probabilities as opposed to class labels are used in the calculation. Because OA, by definition, gives low weight to rare classes, the use of the micro-averaged Dice loss resulted in low accuracies when the model was trained with imbalanced data. However, when the model was trained with balanced training data, the accuracies were very similar to those obtained with macro-Dice. In our data, there was no apparent benefit to using Dice loss instead of weighted CE.

Our experiments provide valuable insight regarding how different loss measures work in practice. Nevertheless, it is important to note that the best or most appropriate loss metric may be case-specific, and therefore, our results may not be applicable to every situation. As a result, the analyst may need to experiment with multiple loss metrics to determine which one is most appropriate for a specific use case.

**Author Contributions:** Conceptualization, S.F. and A.E.M.; formal analysis, S.F. and A.E.M.; writing—original draft preparation, S.F., A.E.M. and T.A.W.; writing—review and editing, S.F., A.E.M. and T.A.W. All authors have read and agreed to the published version of the manuscript.

**Funding:** Funding was provided by the National Science Foundation (Federal Award ID No. 2046059: "CAREER: Mapping Anthropocene Geomorphology with Deep Learning, Big Data Spatial Analytics, and LiDAR"). Any opinions, findings, and conclusions or recommendations expressed in this material are those of the author(s) and do not necessarily reflect the views of the National Science Foundation.

**Data Availability Statement:** The data are available on the West Virginia View website (https://wvview.org) (accessed on 1 January 2024).

**Acknowledgments:** We would like to thank the three anonymous reviewers and the academic editor whose comments strengthened this work. We would also like to thank the originators of the EuroSat dataset used in this study.

**Conflicts of Interest:** The authors declare no conflicts of interest.

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
