# Peer review of "Selecting and Interpreting Multiclass Loss and Accuracy Assessment Metrics for Classifications with Class Imbalance: Guidance and Best Practices"

_remotesensing, doi:10.3390/rs16030533_

Round 1

Reviewer 1 Report

Comments and Suggestions for Authors

The paper makes a significant contribution by addressing the challenges of evaluating classification accuracy in thematic map production, especially in the context of data imbalance. It explores the implications of different metrics and offers guidance on best practices, including the recommendation to avoid redundant accuracy measures and to use weighted cross-entropy loss for training with both balanced and imbalanced data. There are several aspects that should be addressed in the revised version.

The abstract could benefit from a more concise and structured approach, focusing on specific conclusions, recommendations, and research significance for improved clarity and efficiency.

The introduction lacks clear structure and coherence. It would benefit from a more concise and focused approach, directly stating the significance and scope of the study. Additionally, a clearer organization of the content and direct communication of the key conceptual and methodological aspects would enhance its effectiveness.

In the background section, there is some unnecessary content that doesn't directly relate to the main topic (multiclass Loss and imbalance classes). It would be helpful to focus on providing a critical analysis of the literature by synthesizing and integrating the findings, theories, methodologies, and key arguments from the selected sources. Try to identify existing gaps, controversies, and areas of consensus within the literature to provide a more comprehensive and focused overview of the topic.

When describing the methods, ensure that there are sufficient details for readers to repeat the work. Additionally, provide a brief overview of the methodology before delving into specific details, and explicitly state the research objectives and the rationale behind each experiment for enhanced clarity and understanding.

The title "Selecting and Interpreting Multiclass Loss and Accuracy Assessment Metrics for Classifications with Class Imbalance: Guidance and Best Practices" may not fully align with the specific focus on LULC classification models in the paper.

The conclusion should provide a clear and concise summary of the main findings and recommendations, while directly stating the implications of the research for classification model development. Addressing the broader implications and potential limitations of the findings would further enhance the conclusions effectiveness.

Comments on the Quality of English Language

Although there are few grammar errors, the language still needs further refinement

Reviewer 2 Report

Comments and Suggestions for Authors

This paper provides a comprehensive overview of classification accuracy metrics and loss functions for thematic map production, with a focus on issues that arise due to class imbalance. The authors clearly explain terminology, provide insight on redundancy between some metrics, compare macro- and micro-averaging techniques, and offer recommendations. Through experiments on an imbalanced scene classification dataset, this study demonstrate that weighted cross-entropy loss results in the highest accuracy models. The paper also shows that final accuracy statistics should be calculated from population-based confusion matrices, with macro-averages providing a useful integrated measure of class-specific performance.

Some comments and suggestions:

1.Additional experiments comparing focal loss and Tversky loss to weighted cross-entropy could further illuminate choices of loss function.

2.Some class-specific trends get lost in the aggregated evaluation statistics - would be useful to see more analysis.

3.Discussion could better highlight issues that arise when rare classes are also spectrally complex (low sample size and class imbalance confounded).

4.Conclusions strongly positioned weighted cross-entropy as universally best, but noted results may be case specific - recommend softening that conclusion.

Addressing the above comments would provide more nuanced insight and improve clarity.

Comments on the Quality of English Language

N/A

Reviewer 3 Report

Comments and Suggestions for Authors

This article presents some guides and best practices for interpretation of the multiclass loss and selecting the loss function in classification problems in remote sensing. The paper is correctly written and clearly emphasizes the redundancy of some measures. However, it feels that the content is shallow, like a student's exercise with a  known dataset running very few cases in a very limited range of parameters. In my opinion that is not enough for a publication in Remote Sensing.

Although the results are correct, it is less clear what is the relevance of the results given the depth of the study. The work is limited to one network structure varying the loss function among 3/4 different possibilities and varying the training set with balanced/unbalanced cases. Results are presented in two sections, 4.2 about accuracy assesssment metrics and class imbalance and 4.3 about the effect of the imbalance on the training. However 4.2 is mainly about interpreting the results and the conclussions are pretty obvious and can be summarized, as done by the authours, in one paragraph. 

More interesting is the case studied in 4.3 about the impact of the imbalance in the training. However, the study is very limited by the constraint of using simply 3/4 cost functions and leaves out other possibilities like the effect of the network depth to handle the class imbalance (and many, many other possibilities already studied). Only in the case that the only variation possible is the cost function then these results become more relevant. 

At this point I would like ot mention that the problem of class imbalance is a well studied problem in classification in general and also on Deep Learning. Many methods exist to address this problem from the data level the algorithm level or both. That makes this study very limited and the only value that I can see in this type of work is to provide strong evidence to guide other researches or provide clear best practices. This would be the case to the conclussion that weighted CE loss provides the highest accuracy in combination with balanced training data. But again this conclussion seems kind of obvious in the context studied, and it may not hold for some specific problems (e.g. what happens when you can't build a training dataset balanced?). 

For all this, I conclude that the level of this pubalication is not up to the level of the journal and I would recommend rejection. However, this is a decission that shall be taken by the editor rather than me. 

For the manuscript itself, as I said it is well written and I have only these minor comments:

- In Table 1, It is explained the meaning of "+" in the first and second position, but later in the text it appears P++. It is kind of obvious the meaning, but perhaps it would not harm to have it explicitly defined

- In page 5 line 184, it is mentioned "The numerator in Equation 2 is..." I think the reference is wrong and it is meant "The numerator in Equation 3".

- I think there is a typo in Table 7 and the number in the column "Validation-Imbalanced" for teh class "Residential" shall be 40 instead of 400. 

Round 2

Reviewer 1 Report

Comments and Suggestions for Authors

The authors have made revisions and provided explanations based on my comments, and I am in agreement with the publication of this paper.

Reviewer 2 Report

Comments and Suggestions for Authors

I have no more comments.

Comments on the Quality of English Language

n/a